# Effect of the Test Procedure and Thermoplastic Composite Resin Type on the Curved Beam Strength

**DOI:** 10.3390/ma14020352

**Published:** 2021-01-12

**Authors:** Robin Hron, Martin Kadlec, Roman Růžek

**Affiliations:** Materials and Technologies Department, Aviation Division, VZLU-Czech Aerospace Research Centre, 199 05 Prague, Czech Republic; kadlec@vzlu.cz (M.K.); ruzek@vzlu.cz (R.R.)

**Keywords:** composite, thermoplastic, interlaminar strength, polyphenylensulfid, polyetheretherketone, polyaryletherketone, curved beam

## Abstract

The application of thermoplastic composites (TPCs) in aircraft construction is growing. This paper presents a study of the effect of an applied methodology (standards) on out-of-plane interlaminar strength characterization. Additionally, the mechanical behaviour of three carbon fibre-reinforced thermoplastic composites was compared using the curved beam strength test. Data evaluated using different standards gave statistically significantly different results. The study also showed that the relatively new polyaryletherketone (PAEK) composite had significantly better performance than the older and commonly used polyphenylensulfid (PPS) and polyetheretherketone (PEEK). Furthermore, considering the lower processing temperature of PAEK than PEEK, the former material has good potential to be used in serial aerospace production.

## 1. Introduction

More than 95 percent of composites used in the aerospace industry are thermosets [1]. However, the share of high-performance thermoplastic composites (TPCs) in the aeronautical industry is rising year after year even at the expense of these thermosets. It is given by attractive properties such as fracture resistance [2,3,4], formability [5,6], welding [7,8], self-healing possibilities [9,10] and finally recyclability [11]. With regard to modern trends and requirements, we can say that the recyclability of composites belongs (compared to metals) among their weakest aspects. Thanks to thermoplastics, today, we can talk about real recycling of composites. Thermoplastics can be heated and moulded repeatedly without negatively affecting the material’s physical properties. The curing process is completely reversible. These polymers are already polymerized and do not “cure”. They are melt-processable and, due to a lack of cross links, recyclable (reformable) at temperatures above their glass transition temperature. Softening by heating further enables welding of subcomponents. This leads to the elimination of fasteners and adhesives, as is showed in [12,13]. The requested performance of structural TPC parts can be easily achieved by stacking tailored blanks with a combination of the thermoforming process—this is demonstrated, for example, on the thermoplastic rib in [14]. The biggest advantages are that TPCs have a short curing time (compared to thermosets); neither absorb water nor degrade when exposed to moisture; and have excellent fire, smoke and toxicity (FST) properties. The disadvantages of thermoplastics include their higher purchase price. However, the total cost of a component may be less than the thermosetting component due to lower production and storage costs. The most used TPC in aircraft construction is polyfenylensulfid (PPS), and other commonly used thermoplastic matrices are polyetheretherketone (PEEK) and polyetherketoneketone (PEKK). For excellent thermal stability, thermoplastics can be used even when there are higher operating temperatures. The growing production of thermoplastics goes hand in hand with their development. New types of thermoplastic matrices are being introduced into the market. TPCs have been used in advanced airframes, for example, on the horizontal tailplane of AW 169, on the weapon bay doors of F-22, on the rudder of Boeing Phantom Eye, and on the rudder and elevators of G650.

The mechanical behaviour of composite materials is commonly characterized by tensile or compressive strength, by the impact damage and by the environment. There are several material characteristics and methods which can be defined as a barrier for the failure mechanism of composite materials. Two of these characteristics are interlaminar shear strength (ILSS) based on shear loading and interlaminar tensile strength (ILT) based on testing of curved beams. The authors discussed these characteristics previously in References [15,16]. The comparison of ILSS properties on the different types of thermoplastics matrices was formerly studied considering creep and stress relaxation [17], interlaminar shear strength [15] and impact resistance [18]. It could be stated that curved beams better conform to real stress–strain conditions of real curved structures used in composite structures. Additionally, curved beams are sensitive to delamination at locations with high interlaminar stresses. Unfortunately, both the ILSS and ILT values are not readily available (are not included in the material sheets as a standard).

In general, one of the major barriers to accurate failure prediction for polymer-matrix composites is the lack of matrix-dominated material properties, which could be used as a basis for the development of failure criteria [19,20].

The ILT strength generally represents the weakest point of a laminated composite system. At the same time, ILT strength is one of the most difficult material strength properties to characterize [21].

An accurate evaluation of ILT strength is needed to define delamination failure. Delamination is one of the primary failure modes that occur in aerospace composite structures. Currently, the ASTM D6415 [22] and AITM1-0069 [23] curved-beam (CB) methods are standard practices for measurements of ILT strength. Figure 1 shows a typical CB test setup for a 4.95-mm thick fabric PPS material.

The typical failure mode is tensile delamination. Failure starts in the beam radius area at about two-thirds of the thickness. It corresponds to the maximum ILT stress location. Subsequently, delamination quickly propagates through the beam flanges. ASTM D6415 provides equations for ILT strength calculation. Makeev et al. [21] measured the ASTM D6415 CB strength for multiple unidirectional carbon fibre and glass fibre-reinforced epoxy-matrix prepreg tape composites. Based on their experience, the manufacturing process to produce CB coupons with uniform radius and thickness should be preferred. However, it is not possible generally for several practical reasons associated with specific structure design. Additionally, the CB strength data typically exhibit large scatter. For example, Makeev et al. [21] shows that, for 0.26-inch thick CB coupons manufactured from Hexcel IM7/8552 unidirectional tape and cured per manufacturer’s specifications under nominal cure pressure, the average ASTM D6415 ILT strength varies between 68.9 MPa and 82.7 MPa and the coefficient of variation (COV), defined as the ratio of standard deviation to the average value, is usually higher than 20% [21]. The question is whether the ASTM D6415 CB interlaminar strength data, including the large scatter, are coupon-specific. The CB strength is not a coupon-independent material property, suggesting that ASTM D6415 is not an adequate approach to measure the ILT strength of materials. The AITM 1-0069 standard is a very similarly procedure to the ASTM D6415 test and evaluation methods. A comparison of these methods and results evaluated based on defined procedures is discussed by the authors of this paper hereafter.

Another method used for the evaluation of ILT strength is ASTM D7291 [24]. This method applies a tensile force normal to the plane of the composite laminate using adhesively bonded thick metal end-tabs. It was noted in ASTM standard D7291 that thickness strength results using this method will in general not be comparable to ASTM D6415 or AITM 1-0069 since ASTM D7291 subjects a relatively large volume of material to an almost uniform stress field while ASTM D6415 and AITM 1-0069 subject a small volume of material to a nonuniform stress field. It seems that characterization of ILT strength using ASTM D7291 is more representative than ASTM D6415. The reason is the possibility of different failure modes occurring—the failure could occur not only in the composite material but also at the bond lines between the composite and the metal end-tabs. End-tabs are used with the aim of ILT load transfer to the composite.

Formerly, Jackson and Martin [25] studied carbon/epoxy CB specimen configurations to establish a method and specimen for assessing ILT strength. They concluded that specimens with curved geometries include manufacturing problems that cannot be described by flat panels. Failure modes and strengths defined based on curved beam specimens with manufacturing flaws correspond to those in the actual structure with similar flaws. In the case of specimens not containing any significant flaws, a true material property can be defined. Jackson and Martin [25] observed CB strength reduction (up to a factor of four) in low-quality CB specimens containing macroscopic voids detected using fractography analysis. However, it could not explain the large scatter in the strength data observed in high-quality CB specimens. They have not made available the detailed non-destructive inspection (NDI) techniques.

Makeev et al. [21] focused their work on ILT failure and did not address the in-ply transverse tensile failure (matrix ply cracking), which is different from the ILT failure (delamination) discussed by See O’Brien et al. [26,27] for measurement of in-ply transverse tensile material properties. ASTM D6415 significantly underestimates ILT strength in the case where the CB coupon contains porosity even at a low-porosity content. Better values of strength properties can be evaluated after refinement of the ASTM D6415 procedure. This includes measurement of the critical voids in the CB radius area and transition of the defective information into a finite element stress analysis model. The ILT material strength results of the modified CB tests presented by Makeev [21] for the unidirectional IM7/8552 carbon/epoxy tape composite were in excellent agreement with the short-beam tests. The CB tests can be used for assessment of the effects of porosity defects on ILT performance by the refined CB method proposed by Makeev [21]. Hao et al. [28] investigated deformation and strength of CB specimens with various thicknesses and radius–thickness ratios. Strength increased with increases in the thickness and radius–thickness ratio.

The objectives of the presented work are to compare ILT strength of three present-day thermoplastic composite (two commonly used—PPS and PEEK—and one relatively new—polyaryletherketone (PAEK)) materials and to analyse the procedures defined in the ASTM D6415 and AITM 1-0064 standards with the aim of evaluating potentially strength result dissimilarities. Except the matrix type, the effect of the test temperature on ILT strength was evaluated. A comparison using out-of-plane interlaminar strength has been proposed with respect to the fact that interlaminar strength can be two orders of magnitude less than the tensile strength in fibre direction and that even a small load applied in the through thickness direction can lead to the delamination. The basic mechanical and physical properties of the materials used are given in Table 1 and Table 2. The materials were selected on the basis of the manufacturer’s experience and their actual/planned use in the construction of aircrafts.

## 2. Experiment

### 2.1. Material

PPS, PEEK and PAEK thermoplastic polymer melts with T300 3K, 5HS, 280 gm-2 FAW (Fabric Area Weight), 43% RC (Resin Content) (50% by volume) carbon fabric (280 gsm) were compared. The coupons were manufactured by Latecoere Czech Republic, Prague using thermoforming technology. Thermoforming is used to convert a flat consolidated continuous fibre-reinforced laminate into a complex shape with no change in original laminate thickness. The laminates were heated to the required temperature and then quickly formed under ambient pressure with a few minutes dwell time (Figure 2).

Indicative properties of the compared matrices and used carbon laminates with these matrices are shown in the Table 1 and Table 2. Figure 3 graphically illustrates the thermal properties of the compared thermoplastics. By comparing the properties of individual matrices in Table 1, we can see that the PAEK matrix has the highest flexural strength and elongation. On the other hand, it has the lowest compressive strength. The highest compressive strength has a PPS matrix. In tensile strength, the differences are not so large (less than 10%). The greatest differences show thermal properties. The PPS thermoplastics has glass-transition temperature (T_g_) and melt temperature (T_m_) that are significantly lower than the two remaining matrices. These have almost identical T_g_, but PEEK has a T_m_ 38 °C higher than the PAEK matrix. When comparing the fibre-reinforced laminate with PPS, PEEK and PAEK, we find that most properties differ at the minimum. The greatest differences are in the strength properties in the 90° direction and in-plane shear strength; see Table 2.

**Figure 3 materials-14-00352-f003:**
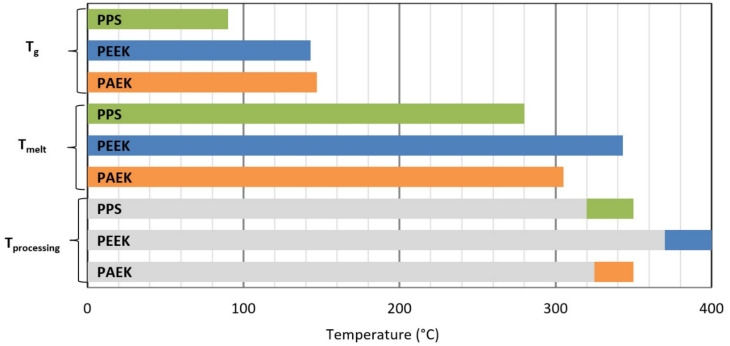
Compared thermal properties of the PPS, PEEK and PAEK matrices [29].

**Table 1 materials-14-00352-t001:** Physical, mechanical and thermal properties of the compared matrices [30,31,32].

Property	PPS	PEEK	PAEK
Specific gravity (g/cm^3^)	1.35	1.3	1.4
T_g_ (°C)	90	143	147
Melt temperature T_m_ (°C)	280	343	305
Moisture absorption (%)	0.02	0.2	0.2
Tensile strength (MPa)	90.3	97.2	95
Tensile modulus (GPa)	3.8	3.59	3.7
Elongation at yield (%)	3	3	4.5
Compression strength (MPa)	148	120	117
Compression modulus (GPa)	3.0	-	-
Flexural strength (MPa)	125	138	141
Flexural modulus (GPa)	3.7	4.1	4.2
Processing temperature (°C)	320–350	370–400	325–350

**Table 2 materials-14-00352-t002:** Mechanical properties of the carbon laminates with the PPS, polyetheretherketone (PEEK) and polyaryletherketone (PAEK) polymer melts [29,30,31].

Property	PPS	PEEK	PAEK
Tensile strength 0° (MPa)	757	776	805
Tensile modulus 0° (GPa)	55.8	56.1	58
Tensile strength 90° (MPa)	754	827	739
Tensile modulus 90° (GPa)	53.8	55.6	59
Compressive strength 0° (MPa)	643	585	628
Compressive modulus 0° (GPa)	51.7	51.6	52
Compressive strength 90° (MPa)	637	595	676
Compressive modulus 90° (GPa)	51.7	49.7	53
In Plane Shear Strength (MPa)	119	155	147
In Plane Shear Modulus (GPa)	4.4	4.5	4.1
Flexural strength 0° (MPa)	1027	-	1040
Flexural modulus 0° (GPa)	60	-	60
Flexural strength 90° (MPa)	831	859	879
Flexural modulus 90° (GPa)	44.8	46.3	48

The samples were divided according to Table 3 as the samples for testing at room temperature (RT) and the samples for testing at a cold temperature of −55 °C (CT). A temperature of −55 °C represents the typical operating temperature in aerospace. Show material properties at this temperature are important for airworthiness.

### 2.2. Material Structure Analysis

Metallographic analysis was performed on the non-tested spare samples for all thermoplastic types. The analysed samples contain only minor porosities (<20 µm in the radius and <200 µm in the flat parts). The void size and quantity were not sufficient to visualize a negative effect on interlaminar strength in cases where the specimens were exposed to a quasi-static loading. No significant deviations between the sets were observed. A typical cross section before testing of PAEK thermoplastic is shown in Figure 4.

### 2.3. Test Method

The objective of the curved beam strength test is to determine the strength characteristics of the composite material in the out-of-plane (z) direction. Radial tensile stress in this direction of the composite (through the thickness of the material) is induced in the curved region of the test specimen when bending is applied. The bending load is applied using a four-point bending fixture with two pairs of cylindrical supports with different span lengths (l_t_ and l_b_).

Before the tests started, a comparison of the two most used test methods was performed (ASTM D6415 [2,3]—the standard test method for measuring the curved-beam strength of fibre-reinforced polymer-matrix composites—and AITM 1-0069 [24]—determination of curved-beam failure loads). This method of loading induces a constant bending moment in the curved region of the specimen. The main motivation was to compare the effect of the spans. Three configurations were prepared: the first was per the ASTM standard, where fix values of span were used (l_t_ = 75 mm and l_b_ = 100 mm); the second configuration used a modified ASTM (ASTM mod) span (l_t_ = 45 mm and l_b_ = 75 mm); and the third was per the AITM standard. In this method, the spans are calculated based on the sample geometry in Equations (1)–(4). Based on these calculations, l_t_ = 26.4 mm and l_b_ = 40.6 mm were set.

(1)$$l_{t}>2\cdot \left(\left((R_{i}+t+\frac{D}{2}\right)\cdot \sin\left(\varphi \right)+\left(\frac{t}{4}+1\right)\cdot \cos\left(\varphi \right)\right)\;\pm 0.5$$

(2)$$l_{t}>2\cdot \left(\left((R_{i}+t+\frac{D}{2}\right)\cdot \sin\left(\varphi \right)+\left(\frac{t}{4}+1\right)\cdot \cos\left(\varphi \right)\right)\;\pm 0.5$$

(3)$$l_{b}>l_{t}+t+10\;\pm 0.5$$

(4)$$l_{b}>l_{t}+t+20\;\pm 0.5$$

In these equations, l_t_ denotes the span of the top fixture, l_b_ is the span of the bottom fixture, R_i_ is the inner radius, t is the thickness of the sample, D is the roller diameter and *φ* is the angle from the horizontal of the sample legs.

Tests were performed on a static load machine Instron 55R1185 (Norwood, MA 02062-2643, 825 University Ave, USA) with an installed load cell with a capacity of ±10 kN and with control system Instron K5178. Recording of the force, displacement and extensometer data was ensured by the software Bluehill 3. The test setup is shown in Figure 5. A test specimen was placed on the bottom cylindrical bars. Then, the extensometer Instron 2620-604 with a base of 50 mm was installed. The extensometer recorded the axial displacement between the upper and lower parts of the fixture. The specimen was loaded by a constant crosshead speed of 2 mm/min and the test ended when the loading rapidly decreased (approximately a 30-percent drop).

### 2.4. Statistical Analysis

Numerous techniques exist for statistical assessment of experiments. For this paper, two types of evaluation were used: (1) T-test for the test method effect where a single factor for two sets is assessed by evaluation of the *p*-value, which is compared to significance level α = 0.05; where for a *p*-value lower than α, the effect is statistically significant; and where the data sets have different mean values and (2) the Taguchi technique of design of experiments (DOE). DOE is the experimental strategy that facilitates the study of multiple factors at different levels. Questions concerning the influence of these factors on the variation of results can only be obtained by performing an analysis of variance (ANOVA).

In this ANOVA design, 2 factors representing both the materials and test temperature were chosen. Three qualitative levels were set for thermoplastic type (A), and two levels were set for the temperature (B). The full factorial experiment made it possible to also investigate the interactions AB of these two factors.

The full model was
(5)$$Y_{ijk}=\mu_{g}+\alpha_{i}+\beta_{j}+{\left(\alpha \beta \right)}_{ij}+\varepsilon_{ijk}$$
(6)$$\varepsilon_{ijk}\sim N\left(0,\sigma^{2}\right)\;i=1,\;2,\;3.;\;j=1,2.;\;k=1,2,\ldots ,6.$$
where *μ_g_* represents a grand mean term common to all observations, *α_i_* is the effect of the *i*th level of A, *β_j_* is the effect of the *j*th level of B, and (*αβ*)*_ij_* is the interaction effect of level *i* of A and level *j* of B combined. Also, a test for normality of residuals *ε* needs to be done.

## 3. Results and Discussion

### 3.1. Test Method Evaluation

A comparison of the methods was performed on the PPS material; see Table 4 and Figure 6. The statistical analysis of data using multiple t-tests showed that the AITM test method has a statistically significant effect (*p*-value = 0.011) when compared to pooled values of the two ASTM methods. It means that the AITM data were different from the other two sets. The results of the ASTM and ASTM mod procedures were not statistically different due to scatter of the data. The results confirmed the general rule that, with shorter distances (span lengths), greater forces are required for samples to fail. In general, it is not always possible to have one geometry available. Dimensions may be based on the actual design of the construction and simulation of a real loading.

For the following experiments, the AITM test method was chosen. The main reason was to take sample geometry into account when setting the test fixture (span length).

### 3.2. Thermoplastic Type and Temperature Evaluation

The highest interlaminar strength was achieved on the set with the PAEK thermoplastic; see Table 5 and Figure 7.

For room temperature (RT), the average interlaminar strength of the PAEK set was to 18% higher than that for PPS and 16% higher than that for the PEEK set. The PAEK set also showed the smallest variance in measured values. Coefficient of variation (CV) for both test temperatures was less than 3%. The greatest variance of the measured values was evaluated on the PPS set (11%).

For cold temperatures (CT), compared to RT, the strength increased by approx. 10% for PPS, by 8% for PEEK and by 6% for PAEK. The values measured on the PAEK sets showed a very small coefficient of variation (less than 3%). For the PPS and PEEK sets, CV was about 10 %.

A series of statistical analyses to compare the sets was performed. First, all measured data were analysed by ANOVA using DOE++ ReliaSoft (Version 1.0.7; ReliaSoft Corporation, MI, USA). The purpose was to evaluate the main factors and their effects on the results using a general full factorial design with multiple level factors—temperature and thermoplastic type. The Anderson–Darling test of residuals for model (5) proved the normality. The results of analysis for each factor in model (5) are shown in Table 6 in following order: A: thermoplastic type, B: temperature and AB: interaction interactions between both factors. Statistically significant effects of both testing temperature (B) and thermoplastic type (A) on measured strength values were found (*p*-values < 0.05). The interaction of effects (AB) was not proven. In a graphical form, the results are illustrated in Figure 8.

A complete evaluation of the differences for each parameter is shown in Table 7. *t*-tests applied for individual sets revealed that the PAEK interlaminar strength was higher than that for both the PEEK and PPS sets for both temperatures. No difference was proven between the PPS and PEEK sets. Cold conditions at −55 °C increased interlaminar strength in all investigated cases compared with RT conditions.

For better failure identification, the edges of the sample were painted in white colour. Valid failure occurred for all samples: a delamination in curvature occurred. Figure 9 shows the PPS-CT set failure modes. This set has a relatively high coefficient of variation (<10%). It can be seen from the figure that, at the lowest values, the failure occurred locally, and, in this set, specifically in the middle (sample 9) or at the outer radius (sample 10). For the other samples (6–8), the failure was over the entire thickness of the test sample. Figure 10 shows the PAEK-CT set samples. This set had a small coefficient of variation (2.8%), and similar failures were noted. Localization of the failures could be caused by clustering of the porosity in one place, by a manufacturing defect, or as a natural property of the material.

## 4. Conclusions

Three methods for interlaminar strength evaluation were analysed. Data evaluated using the AITM standard give significantly different results from the data based on the ASTM procedures. This implies that results from different methods cannot be directly compared and used for numerical analyses.

On the basis of statistical evaluation of three different composite materials, it can be stated that the choice of thermoplastic type can have a significant effect on the values of interlaminar strength. The values measured on the PAEK samples were statistically significantly higher than those measured on the PPS and PEEK samples (valid for both test temperatures).

The metallographic analysis performed on the samples showed similar homogeneity of all investigated materials, and the void size and quantity were not sufficient for a negative effect on interlaminar strength.

Cold conditions at −55 °C increased interlaminar strength in all investigated cases compared with RT conditions. The PAEK material gives significantly higher interlaminar shear strength in comparison with the PPS and PEEK materials. Additionally, the PAEK material has the lowest scatter in measured data. The PAEK thermoplastic material is a new composite material planned for application in future aircrafts.

Further tests on the coupon with TPC matrix are planned for the future in order to expand the material database. Based on these tests, a material will be selected that could replace the thermosetting materials used in aircraft structures for interior panels.

## Figures and Tables

**Figure 1 materials-14-00352-f001:**
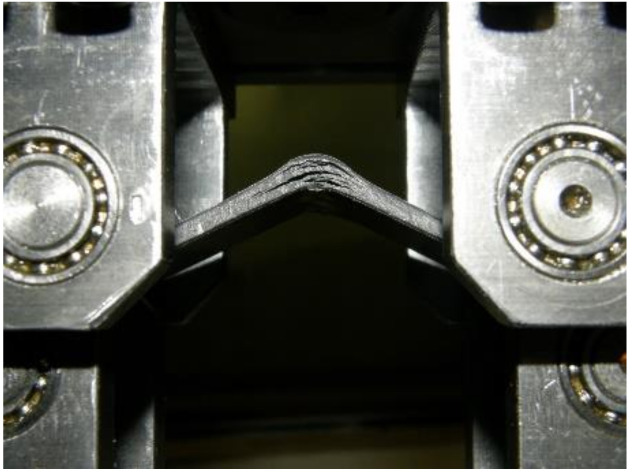
Curved-beam test setup and delamination failure of a polyphenylensulfid (PPS) specimen.

**Figure 2 materials-14-00352-f002:**
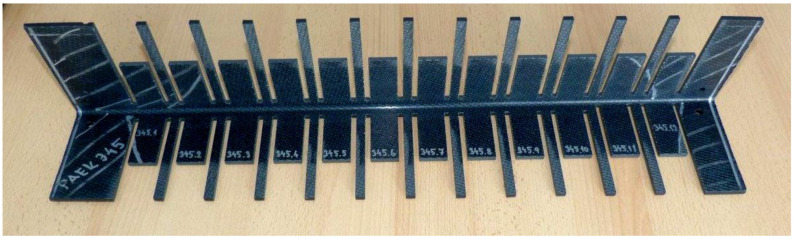
Curved-beam strength samples.

**Figure 4 materials-14-00352-f004:**
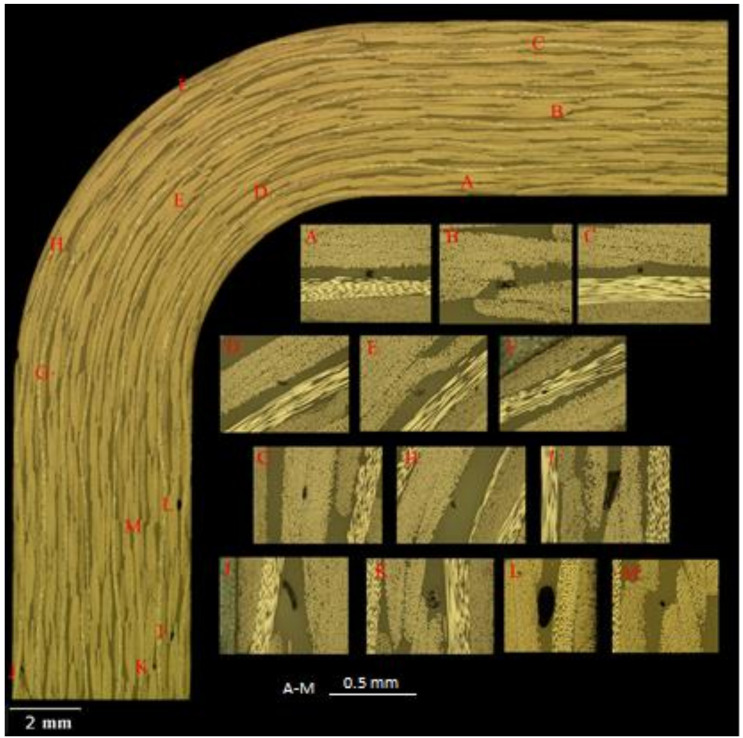
Example of the cross section before testing: PAEK.

**Figure 5 materials-14-00352-f005:**
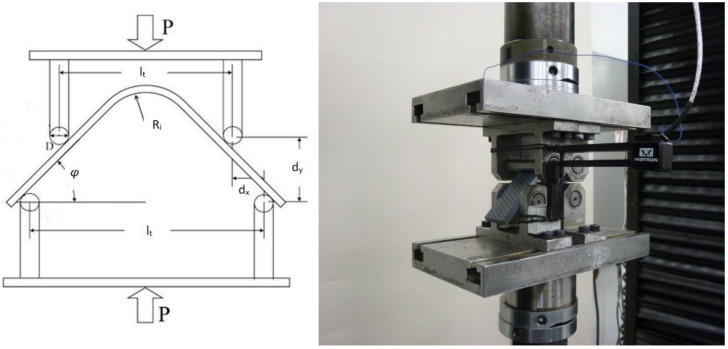
Curved-beam strength test.

**Figure 6 materials-14-00352-f006:**
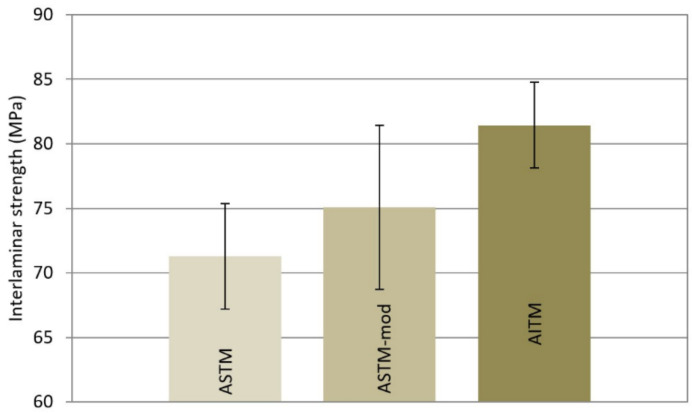
Measured interlaminar strength, σ_r_ (MPa): the test method comparison using statistic evaluation showed a significant difference of the AITM method.

**Figure 7 materials-14-00352-f007:**
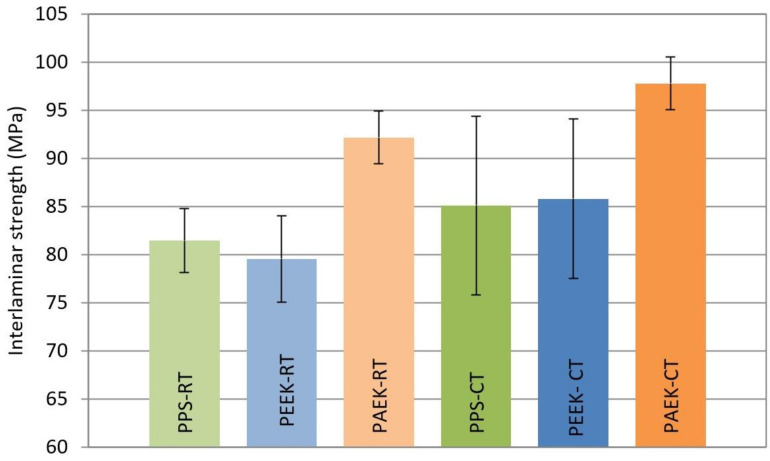
Comparison of the interlaminar strength for 6 data sets, *p*-value 0.05.

**Figure 8 materials-14-00352-f008:**
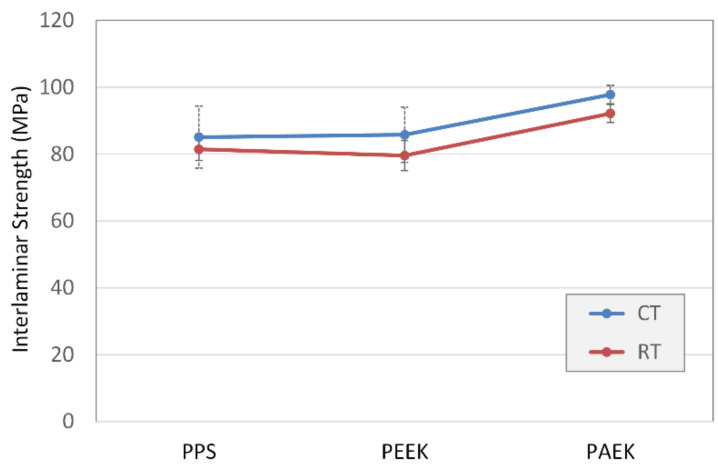
Graphical comparison of the interlaminar strength results: temperature and thermoplastic-type influence.

**Figure 9 materials-14-00352-f009:**
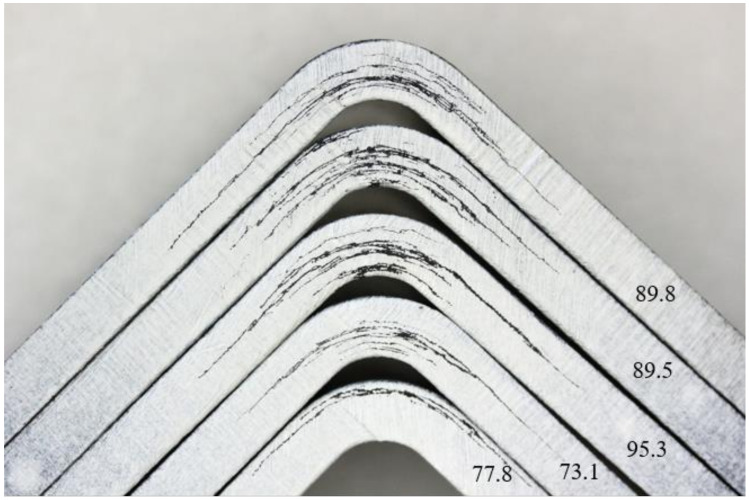
Examples of a typical failure mode for set PPS-CT: the set with the highest coefficient of variation, with the values given in the figure being strengths in MPa.

**Figure 10 materials-14-00352-f010:**
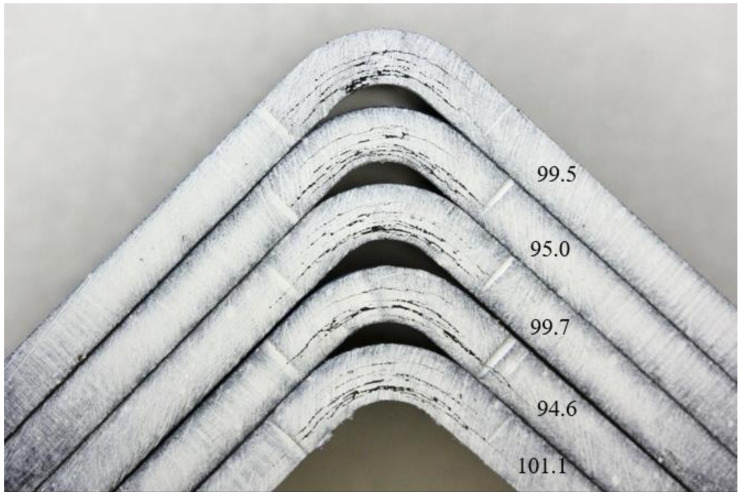
Examples of a typical failure mode for set PAEK-CT: the set with the lowest coefficient of variation, with the values given in the figure being strengths in MPa.

**Table 3 materials-14-00352-t003:** Overview of the tested sets.

Set (Resin)	Fabric	Lay-Up	Ø Width (mm)	Ø Thickness (mm)	Ø α (Deg)	Number of Samples	
						RT	CT
PEEK	T300JB 3K, 5HS, 280 gsm FAW, 42% RC (50% BV)	[(0,90)/(±45)]_4_/(0,90)	25.31	2.78	91.0	6	5
PPS	T300 3K, 5HS, 280 gsm FAW, 43% RC (50% BV)	[[(0,90)/(±45)]_4_]	25.14	4.95	89.5	5	5
PAEK	T300JB 3K, 5HS, 277 gsm FAW, 42% RC (50% BV)	[[(0,90)/(±45)]_4_]	25.24	4.65	90.7	5	5

**Table 4 materials-14-00352-t004:** Comparison test method influence on interlaminar strength, σ_r_ (MPa).

σ_r_ (MPa)	PPS		
	ASTM	ASTM Mod	AITM
Mean	71.3	75.1	81.4
S.D.	4.09	6.36	3.33
C.V.	5.73	8.47	4.09
Min.	65.4	68.9	84.7
Max.	78.8	85.6	77.4

**Table 5 materials-14-00352-t005:** Measured interlaminar strength, *σ*_r_ (MPa).

-	**Interlaminar Strength (MPa)**					
	**RT**				**CT**	
	**PPS**	**PEEK**	**PAEK**	**PPS**	**PEEK**	**PAEK**
	83.6	79.8	91.8	89.8	87.3	99.5
	84.7	86.4	88.3	89.5	78.2	95.0
	80.0	73.3	92.0	95.3	78.1	99.7
	63.5	78.6	96.0	73.1	87.0	94.6
	77.4	82.3	92.7	77.8	98.2	100.1
	-	77.0	-	-	-	-
**Mean**	**77.9**	**79.6**	**92.2**	**85.1**	**85.8**	**97.8**
S.D.	8.52	4.49	2.73	9.26	8.29	2.75
C.V.	10.95	5.65	2.97	10.89	9.66	2.81
Min.	63.5	73.3	88.3	73.1	78.1	94.6
Max.	84.7	86.4	96.0	95.3	98.2	100.1

**Table 6 materials-14-00352-t006:** ANOVA analysis of thermoplastic type and temperature influence on data results.

Source of Variation	Degrees of Freedom	Sum of Squares (Partial)	Mean Squares (Partial)	F Ratio	*p*-Value
Model	5	1468.419	293.6838	6.8664	0.0004
A: thermoplastic type	2	1128.575	564.2875	13.1932	0.0001
B: Temperature	1	312.3451	312.3451	7.3027	0.0122
AB	2	3.4559	1.7279	0.0404	0.9605
Residual	25	1069.2783	42.7711		
Pure Error	25	1069.2783	42.7711		
Total	30	2537.6973			

**Table 7 materials-14-00352-t007:** Statistical comparison of individual files using *t*-tests. D, files are different; ND, files are not different.

-		RT				CT	
		PPS	PEEK	PAEK	PPS	PEEK	PAEK
RT	PPS	-	ND	D	ND	ND	D
	PEEK	ND	-	D	ND	ND	D
	PAEK	D	D	-	ND	ND	D
CT	PPS	ND	ND	ND	-	ND	D
	PEEK	ND	ND	ND	ND	-	D
	PAEK	D	D	D	D	D	-

## Data Availability

Data sharing is not applicable to this article.

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
