# Peer review of "Effect of the Test Procedure and Thermoplastic Composite Resin Type on the Curved Beam Strength"

_materials, 2021, doi:10.3390/ma14020352_

Round 1

Reviewer 1 Report

This paper is, from a technical point of view, very good, needed and useful. This is a timely manuscript that presents a set of interesting and useful results. The research is properly framed and clearly presented.
Only two notes before publication:
a) Please revise lines 175 e 176;
b) Authors use exhaustively the term “statistically significantly”: Please revise

Author Response

Dear reviewer, thank you for your comments. With the wish of good health.

Robin Hron

Reviewer 2 Report

Reviewer Comments “Effect of the test procedure and thermoplastic 2 composite resin type on the curved beam strength”

The authors present a research study where they first evaluate different standards to assess the curved beam strength of composite materials. They find that different standards may lead to varying results. In addition, they compare the materials properties of three different PMC. In detail, they wanted to see if thermoplastic polymers could also serve as a matrix material for fiber-reinforced composites, especially for the aerospace industry.

Overall, the manuscript is well written. However, I have some suggestions that would improve this work.

  • As mentioned in the introduction, here are thermoplastic polymers investigated. It should be noted that these polymers are already polymerized and do not ‘cure’. They are melt- processable and due to a lack of cross-links recyclable (reformable) at temperatures above their glass transition temperature.
  • In addition, I would not call them resin. A resin is usually a non-cured monomer solution (or pre-polymer solution) that after curing/crosslinking results in a thermoset polymer. It would be more precise to rename it to thermoplastic, polymer, or polymer melt.
  • In the introduction, you could give more information on PMC, different filler types, geometries, and the processing (fiber, particles, mats, direction, continuous/discontinuous etc). What specific filler material is investigated here and what type of composite is used and why? This only becomes clear after reading further, but should already be introduced here.
  • I noticed that the three composites had slightly different fillers (T300 vs 300JB, 280 vs 277 gsm, 42 vs 43% RC) and different thicknesses. Could you comment on this? It seems like this adds more variables that should be considered when comparing the results.
  • When comparing the results from table 4, you could add a discussion explaining why these results make sense based on the different geometries. With shorter distances (span length), higher forces are needed for samples to fail.
  • Line 206. The statistical analysis comes out of the blue here with not enough detail. I would suggest adding a section to the experimental part where you describe in detail what statistical methods were used and how.
  • Figure 6. When denoting statistical significance, it might be helpful to indicate that in the graphic as well. Usually, stars are used, details (p-value) need to be added to the figure caption.
  • What is the rationale for -55C? Please add a paragraph to the manuscript.
  • Table 5 and Figure 7: You could combine both by showing a box whisker plot instead. That would be more informative to the reader and easier to capture. Here, you could again add the stars above to indicate the statistical difference.
  • Line 226. And again, having a section in the experimental where the statistical methods are details would be better.
  • Table 6. Here, you mention ANOVA for the first time. Se comments before. I think that the table itself could be moved to the supplemental information. It seems not to be essential.
  • Table 7. It would be easier for the reader if D, ND would be color-coded (green/red for example). That would make it easier to capture.
  • Line 246 and following. Could you discuss why the properties at lower temperatures are different? What is the molecular mechanism behind this?
  • Figs 9 and 10 would benefit if you would add the sample numbers that are mentioned in the text to the images to better identify them.
  • Conclusion: What are future directions?

Author Response

Dear evaluator, thank you for your comments, suggestions and opinions. I respect them very much.

With good health wishes.

Robin Hron

Reviewer 3 Report

The work entitled " Effect of the test procedure and thermoplastic composite resin type on the curved beam strength" is can be beneficial for the composite community. Please accept the manuscript after addressing the following comments-

  1. Page 4, line 138: Please include a specific pressure value as it is well known that pressure variance may introduce various free volume in the samples.
  2. Figure 4: if I am not wrong, each individual image (A-M) is not in the same resolution as the given 2mm scale. Please include a scale bar for each image.
  3. Page 7, line 175/176: please correct typo/ symbol/ reference.

Author Response

Dear evaluator, thank you for your comments.

With good health wishes.

Robin Hron

Round 2

Reviewer 2 Report

Thanks for the revision and for taking my comments into account. My major concerns have been addressed. I still think that it would be good to include one small section that describes the statistical methods at the end of the Experimental part, as separate section 2.4 after test methods. This would not distract the reader and is valuable information that helps the reader to put the results into the right context.

Author Response

Dear Reviewer,

I have to admit, you're right. In the 2. Experiment chapter, my colleagues and I added subchapter 2.4 Statistical analysis.

Have a nice day.

R.Hron